# The antiangiogenic peptide VIAN-c4551 inhibits lung melanoma metastasis in mice by reducing pulmonary vascular permeability

**Alma Lorena Perez[1], Magdalena Zamora[1], Manuel Bahena[1], Regina Aramburo-Williams[1], Elva Adan-Castro[1], Daniela Granados-Carrasco[1], Thomas Bertsch[2], Jakob Triebel** [2]**, Gonzalo Martinez de la Escalera[1], Juan Pablo Robles[1,3‡], Carmen Clapp[1‡***

1 Instituto de Neurobiología, Universidad Nacional Autónoma de México (UNAM), Querétaro, México,
2 Institute for Clinical Chemistry, Laboratory Medicine and Transfusion Medicine, Nuremberg General Hospital & Paracelsus Medical University, Nuremberg, Germany, 3 VIAN Therapeutics, San Francisco, California, United States of America

‡ These authors share senior authorship on this work.

* clapp@unam.mx

## Abstract

### Introduction

Cancer cells drive the increase in vascular permeability mediating tumor cell extravasation and metastatic seeding. VIAN-c4551, an antiangiogenic peptide analog of vasoinhibin, inhibits the growth and vascularization of melanoma tumors in mice. Because VIAN-c4551 is a potent inhibitor of vascular permeability, we evaluated whether its antitumor action extended to a reduction in metastasis generation.

### Methods

Circulating levels of vascular endothelial growth factor (VEGF), lung vascular permeability, melanoma cell extravasation, and melanoma pulmonary nodules were assessed in C57BL/6J mice intravenously inoculated with murine melanoma B16-F10 cells after acute treatment with VIAN-c4551. VEGF levels, transendothelial electrical resistance, and transendothelial migration in cocultures of B16-F10 cells and endothelial cell monolayers supported the findings.

### Results

B16-F10 cells increased circulating VEGF levels and elevated lung vascular permeability 2 hours after inoculation. VIAN-c4551 prevented enhanced vascular permeability and reduced melanoma cell extravasation after 2 hours and the number and size of macroscopic and microscopic melanoma tumors in lungs after 17 days. In vitro, VIAN-c4551 suppressed the B16-F10 cell-induced and VEGF mediated increase in endothelial cell monolayer permeability and the transendothelial migration of B16-F10

**Data availability statement:** All relevant data are within the paper and its Supporting Information files

**Funding:** This work was supported by the Dirección General de Asuntos del Personal Académico (DGAPA) Universidad Nacional Autónoma de México (UNAM) (Grant IN202424) to CC. Alma Lorena Perez is a doctoral student from the 'Programa de Doctorado en Ciencias Biomédicas, Universidad Nacional Autónoma de México (UNAM)' and has received fellowship 788879 from Consejo Nacional de Humanidades, Ciencia y Tecnología (CONAHCYT). MZ, EA-C, and JPR are post-doctoral fellows from CONAHCYT. The funders had no role in study design, data collection and analysis, decision to publish, or preparation of the manuscript.

**Competing interests:** JPR, MZ, TB, JT, GME, and CC are inventors of a submitted patent application (WO/2021/098996), which is owned by the Universidad Nacional Autónoma de México (UNAM) and JT. JPR is the CEO and founder of VIAN Therapeutics Inc. MZ and CC are consultants for VIAN Therapeutics. Inc. This does not alter our adherence to PLOS ONE policies on sharing data and materials.

cells. No detrimental effect of VIAN-c4551 was observed on hematological, biochemical, and histological parameters after its intravenous administration in mice for 14 days.

## Conclusions

These findings support the inhibition of distant vascular permeability for the prevention of tumor metastasis and unveil the anti-vascular permeability factor VIAN-c4551 as a potential and safe therapeutic drug able to prevent metastasis generation by lowering the extravasation of melanoma cells.

## Introduction

Metastasis is the major cause of cancer death, yet metastatic activity is not the main end point of current anti-cancer treatments, and the process remains elusive [1]. A critical step in early metastasis is the exiting of tumor cells from the circulation at distant sites. Extravasation depends on the ability of tumor cells to transmigrate through the vessel wall. Tumor cells can disrupt the vascular barrier at metastatic sites by secreting vascular permeability factors [2], including VEGF, the most powerful vascular permeability stimulator [3]. The development of drugs targeting the extravasation of tumor cells offers a hopeful approach to prevent or delay the metastatic outgrowth of cancer.

VIAN-c4551 is a promising anti-cancer drug. It is a highly potent antiangiogenic cyclic heptapeptide analog of vasoinhibin [4], an endogenous antiangiogenic and vascular permeability inhibitor with significant therapeutic potential [5]. Vasoinhibin inhibits endothelial cell proliferation, migration, survival, and permeability in response to VEGF and other angiogenic and vascular permeability stimulators (basic fibroblast growth factor, bradykinin, interleukin 1β) [4,5]. Vasoinhibin helps restrict physiological angiogenesis in the retina and cartilage [6,7] and inhibits pathological angiogenesis and vasopermeability in experimental models of vasoproliferative retinopathies [8], inflammatory arthritis [9], peripartum cardiomyopathy [10], preeclampsia [11] and cancer [12]. Vasoinhibin gene transfer in prostate [13], colon [14], mammary gland [15] and melanoma [16,17] cancer cells reduce primary tumor growth and neovascularization. Furthermore, vasoinhibin gene therapy delivered two days after the intravenous inoculation of melanoma cells reduces the number and size of lung melanoma nodules [16] indicating its ability to inhibit the growth of metastasis after the engraftment of tumor cells in the metastatic region but there is no information of an action before metastatic spreading.

However, the direct use of vasoinhibin as an anti-cancer therapeutic agent is hampered by difficulties in its recombinant production [18]. These difficulties have been overcome by the development of VIAN-c4551, a vasoinhibin analog with improved pharmacological properties that conserves the efficacy and potency of vasoinhibin [4]. VIAN-c4551 inhibits VEGF-induced proliferation and permeability of endothelial cells with a potency like vasoinhibin (IC$_{50}$ = 150 pM) and is orally active to inhibit

melanoma tumor growth and vascularization in mice [4,19]. Because extravasation is a pivotal step of metastasis involving tumor vasopermeability factors such as VEGF [2,3] and VIAN-c4551 is a potent inhibitor of VEGF-induced vasopermeability [4,19], we aimed to investigate whether the anti-tumor action of VIAN-c4551 extended to the inhibition of the VEGF-mediated vascular permeability causing the extravasation and metastatic spread of melanoma cells.

## Methods

### Reagents

VIAN-c4551 (>95% pure) was synthesized by GenScript (Piscataway, NJ, USA).

### Cell culture

The mouse melanoma B16-F10 cell line (CRL-6475, ATCC, Manassas, VA, USA) expressing (B16-F10-GFP) or not the green fluorescent protein, the NIH/3T3 mouse embryonic fibroblast cell line (CRL-1658, ATCC), the human breast cancer cell lines MCF7 (HTB-22, ATCC) and MDA-MB-231 (HTB-26, ATCC), and the bovine umbilical vein endothelial cell line (BUVEC-E6E7), generated and characterized as reported [20], were cultured in high glucose Dulbecco's Modified Eagle Medium (DMEM) (12100–038, GIBCO, Thermo Fisher Scientific, Waltham, MA, USA) supplemented with 10% fetal bovine serum (FBS) (26140–079, GIBCO) and 100 U/ml penicillin-streptomycin (L0022, Biowest, Bradenton, FL, USA). Human umbilical vein endothelial cells (HUVEC) were obtained [21] and cultured in F12K medium (21127–022, GIBCO) supplemented with 20% FBS, 100 µg/ml heparin (H3149, Sigma-Aldrich, Saint Louis, MO, USA), 25 µg/ml ECGS (356006, Corning Inc., Corning, NY, USA), and antibiotics.

### Vascular permeability in vitro

BUVEC-E6E7 were seeded on 6.5 mm transwell clear polyester membrane inserts (Corning) with pore sizes of 0.4 µm at an initial density of $7.2 \times 10^3$ cells per well (30% confluency). After approximately 96 hours, transendothelial electrical resistance (TEER) measured by the EVOM2 Epithelial Voltohmmeter (World Precision Instruments, Sarasota, FL) stabilized around 55 $\Omega.cm^2$ (reflecting a confluent monolayer) and treatments started. They included vehicle (DMEM 10% FBS), 100 nM VIAN-c4551, or 2 µg anti-VEGF (Ranibizumab, Lucentis®, Novartis, Basel, Switzerland) for 1 hour before adding B16-F10 cells ($3.5 \times 10^4$ cells in 100 µl per well) or applying 100 µl/well of medium conditioned by B16-F10 cells (B16-F10-CM) or medium conditioned by 3T3 cells (3T3-CM) to the luminal side. TEER was measured over a 6-hour period, with high TEER values indicating a tight, impermeable barrier, and lower values indicating increased permeability. In other experiments, HUVEC were seeded at an initial density of $5 \times 10^3$ cells/well (40% confluency) on 6.5 mm transwell with 0.4 µm pores. The TEER stabilized around 55 $\Omega.cm^2$ after approximately 96 hours, and the monolayers were then treated with vehicle or 100 nM VIAN-c4551 for 1 hour followed by the addition of 100 µl/well of B16-F10-CM, or medium conditioned either by MCF-7 cells (MCF-7-CM) or MDA-MB-231 cells (MDA-MB-231-CM). TEER was measured over a 2-hour period. All CM were from cells grown to 80% confluency for 48 h. VEGF levels in the CM of B16-F10 cells treated or not with 100 nM VIAN-c4551 were measured by the enzyme-linked immunosorbent assay (ELISA, Quantikine mouse VEGF kit #MMV00, R&D System, Minneapolis, MN, USA).

### Actin distribution

A previously described method was used [22]. BUVEC-E6E7 were seeded on coverslips placed in 24-well plates and grown to confluence. When the monolayer was completely formed, cells were treated for 1 hour with 100 nM VIAN-c4551, 2 µg anti-VEGF (Ranibizumab, Lucentis®, Novartis) or PBS followed by the addition of 100 µl B16-F10-CM or vehicle (PBS) for 1 hour. Cells were then washed, fixed with 4% paraformaldehyde for 20 minutes, permeabilized with 0.1% Triton X-100 for 15 minutes, stained for actin with 160 nM rhodamine–phalloidin (R415, Thermo Fisher Scientific) for 1 hour in

darkness, and counterstained with 5 µg/ml Hoechst 33342 (B2261, Sigma-Aldrich). Finally, the coverslips were washed, mounted, and observed under a fluorescent microscope (Olympus BX60, Tokyo, Japan).

## Transendothelial migration assay

BUVEC-E6E7 ($7.2 \times 10^3$) were seeded on 6.5 mm transwell membrane inserts with 8 µm pores coated with 0.38 mg/ml matrigel (354234, Corning). Upon confluency (55 $\Omega.cm^2$, revealed by the EVOM2 resistance tester) monolayers were treated for 1 hour with 100 nM VIAN-c4551 into the upper (luminal) chamber followed by the addition of fluorescent B16-F10-GFP cells ($3.5 \times 10^4$ cells per well). Conditioned medium of B16-F10 cells containing 10% FBS was added into lower (abluminal) chamber to serve as chemoattractant. After incubation for 16 hours, cells in the luminal side were washed and migrating cells in the abluminal side were fixed, observed by an inverted fluorescent microscope (Olympus IX51), and quantified using the Image Pro Plus software (Version 7, Media Cybernetics Inc., Rockville, MD, USA).

## Animals

Adult female C57BL/6J mice (8–12 weeks) were selected based on their use in preceding studies showing the inhibition of melanoma tumor growth and neovascularization by vasoinhibin [16] and VIAN-c4551 [4]. Mice were housed under standard laboratory conditions (22°C; 12h/12h light/dark cycle; free access to food and water). Experiments were approved by the Bioethics Committee of the Institute of Neurobiology of the National University of Mexico (UNAM) according to the US National Research Council's Guide for the Care and Use of Laboratory Animals (Eighth Edition, National Academy Press, Washington, D.C., USA). B16-F10 cells were trypsinized, washed and resuspended in PBS. Restrained, non-anesthetized mice were injected into the lateral tail vein with vehicle (saline) or 1 mg/kg body weight (b.w.) VIAN-c4551, the effective dose characterized in a preceding study [4]. Thirty minutes later, $2 \times 10^5$ B16-F10 or B16-F10-GFP cells resuspended in 100 µl PBS or 100 µl PBS were injected into the tail vein. Mice in control and treatment groups were selected at random and identified with ear tags to minimize potential confounders. Groups were divided to either collect blood, evaluate pulmonary vascular permeability, or quantify melanoma cell extravasation 2, 4 or 8 hours after B16-F10 cell intravenous (i.v.) inoculation. Other groups were euthanized 17 days after B16-F10 cell injection when their lungs were harvested to evaluate macroscopic and microscopic melanoma tumors. Animals were monitored every other day and showed no apparent signs of discomfort, loss of body weight, or impaired quality of life that merited early euthanasia. At endpoint times (2–8 hours or 17 days post B16-F10 cell inoculation), mice were anesthetized (60% ketamine/40% xylazine; 1µl/g b.w.) before perfusion and euthanized by $CO_2$ inhalation or an overdose of ketamine/xylazine and decapitation. A total of 132 animals were used, no animals died or had to be excluded. Investigators trained in animal care and handling performed all experiments and were blind to treatment assignments when evaluating outcomes. Sample size was defined based on reliable differences.

## Pulmonary vascular permeability

Vascular permeability was assessed by the extravasation of albumin stained by the Evans blue dye as previously described [23] with some modifications. Briefly, the Evans blue dye (50 µl, 45 mg/kg; E2129, Sigma-Aldrich) was i.v. injected after B16-F10 cell inoculation 1 hour before perfusion. Five hundred µl of blood was withdrawn from the heart to measure Evans blue concentration in plasma, and mice were then perfused for 2 minutes via the right ventricle with 70 ml PBS (pH 3.5 at 37°C). Lungs were dried at 72°C for 24 hours and the Evans blue dye extracted by incubating each lung in 300 µl formamide (F7503, Sigma-Aldrich) for 18 hours at 72°C. Absorbance was measured in the supernatant at 620 nm using the Varioskan Flash spectrophotometer (Thermo Fisher Scientific). The dye concentration in the extracts was calculated using a standard curve of Evans blue in formamide and normalized to the lung and body weight and to the Evans blue concentration in plasma. VEGF levels in serum were measured by ELISA (Quantikine mouse VEGF kit #MMV00, R&D System).

### Lung metastasis melanoma model

Lungs were harvested and placed in Fekete's solution as reported [24]. Tumor nodules were evaluated macroscopically on the lung surface with a stereoscope (Leica Zoom 2000, Leica Biosystems, Nussloch, Germany) and their number and size quantified using the ImageProPlus analysis software by 2 independent operators blind to treatment. Samples were fixed in 10% formalin for 48 hours, dehydrated in ethanol, and embedded in paraffin. Seven-µm-thick paraffin sections from lungs were deparaffinized with xylol, rehydrated in graded alcohol series, stained with Harris's hematoxylin and eosin solution, and digitalized using Aperio Image ScanScope (Leica) and the number and area of microscopic nodules were quantified with the ImageProPlus analysis software.

### Lung melanoma cell extravasation

Mice were perfused with 10 ml PBS for 10 minutes (1ml/minute) via the right ventricle with PBS (pH 7.4 at 37°C) to remove nonadhered melanoma cells. The left and right lungs were processed for RT-qPCR and histological evaluation, respectively. RNA was isolated using TRIzol (15596018, Invitrogen, Thermo Fisher Scientific) and retrotranscribed with the high-capacity cDNA reverse transcription kit (4368813 Applied Biosystems, Thermo Fisher Scientific). Polymerase chain reaction (PCR) products were obtained and quantified using Maxima SYBR Green qPCR Master Mix (K0223, Thermo Fisher Scientific) in a final reaction containing 20 ng of cDNA and 0.5 µM of each of the following primer pairs for murine genes: GFP fwd (5'-AAGTCGTGCTGCTTCATGTG-3'), GFP rev (5'-CAAGCTGACCCTGAAGTTCA-3'), GAPDH fwd (5'-GAAGGTCGGTGTGAACGGATT-3') and GAPDH rev (5'- TGACTGTGCCGTTGAATTTG-3'). Amplification consisted of 40 cycles of 10 seconds at 95 °C, 30 seconds at the annealing temperature of each primer pair, and 30 seconds at 72 °C. The mRNA expression levels were calculated by the $2^{-\Delta\Delta CT}$ method. The right lung was fixed in 4% paraformaldehyde for 24 hours, cryoprotected with 4% sucrose (4072-05, JT Baker, Phillipsburg, NJ, USA) for 48 hours, frozen and cryosectioned (20 µm) (Leica CM1850). Sections were observed under a fluorescent microscope (Olympus BX60) to visualize extravasated B16-F10-GFP cells.

### Toxicity evaluation

A sub-acute toxicity analysis was performed in mice treated with VIAN-c4551. Briefly, six female mice (6-week-old) per group were daily injected i.v. into the lateral tail vein with vehicle (saline) or with a single dose of VIAN-c4551 (1 or 10 mg/kg) for 14 days. On the following day, mice were fasted for 4 hours, weighed, euthanized by $CO_2$ inhalation, and the blood, withdrawn by cardiac puncture was placed in heparin microtubes for the evaluation of hematological and biochemical parameters. Heart, lungs, liver, spleen, and both kidneys were dissected, weighed and collected for histopathological examination. Tissues were fixed in buffered formalin (10% formaldehyde) and processed by independent certified veterinary pathologists (Biocel Laboratorio Veterinario Queretaro, Mexico).

### Statistical analysis

The overall significance threshold was set at $P < 0.05$. The unpaired two-tailed t-test was used for two-group comparisons and one-way or two-way ANOVA for more than two groups. After ANOVA, Tukey, Dunnett and Sidak post hoc tests were performed using GraphPad Prism version 10.4.0 for macOS (GraphPad Software, San Diego, CA, USA).

## Results

### VIAN-c4551 reduces the permeability of pulmonary vessels stimulated by B16-F10 cells

Tumor cells stimulate the vascular permeability needed for their extravasation at metastatic sites [2,25]. To evaluate melanoma cell-induced vascular leakage, mice were injected i.v. with B16-F10 cells and vascular permeability determined in lungs by the extravasation of Evans blue stained albumin at different times post-B16-F10 cell inoculation (Fig 1a). Lung

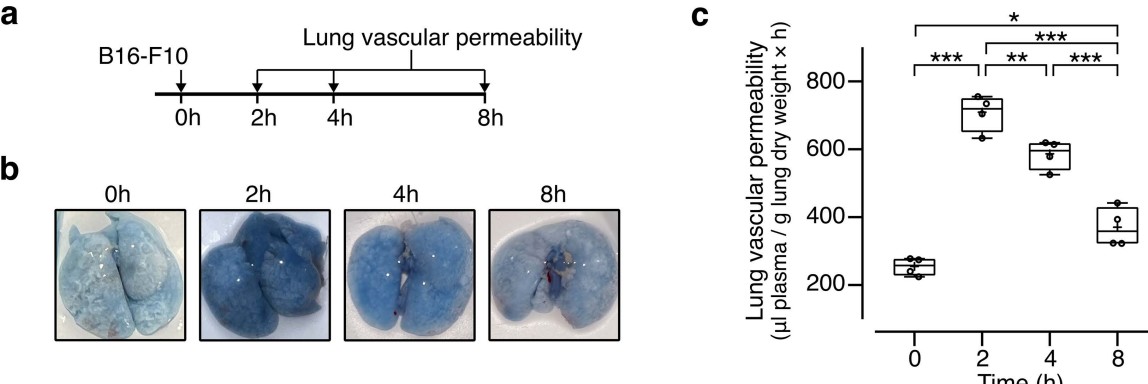

**Fig 1. B16-F10 cells stimulate lung vascular permeability. (a)** Timeline of the evaluation of pulmonary vascular permeability after intravenous inoculation of B16-F10 cells. **(b)** Photographs of representative lungs showing the accumulation of Evans blue-stained albumin at different times post-B16-F10 cell delivery. **(c)** Quantification of Evans blue-stained albumin as index of lung vascular permeability. Data from 2 independent experiments (n=4) is graphed using a box and a whisker plot; the box frames the interquartile range, the horizontal line indicates the median, and the whiskers the min and max values. The mean is indicated by +. *P=0.02, **P=0.01, ***P<0.001 (One-way ANOVA, Tukey's multiple comparison test).

vessels responded to circulating B16-F10 cells by the accumulation of the Evans blue tracer (Fig 1b) with the increase in vascular permeability being highest and lowest at 2 hours and 8 hours post-treatment, respectively (Fig 1b and c). The timing of lowest leakage (8 hours) coincided with the reported time at which most melanoma cells have extravasated following their systemic inoculation [26] and suggested 2 hours post-tumor cell injection as the window of opportunity for modifying the exit of tumor cells from the intravascular to the extravascular compartment.

VIAN-c4551 or vehicle (saline) was i.v. injected 30 minutes before the systemic inoculation of B16-F10 cells or the i.v. injection of PBS followed by the evaluation of lung vascular leakage or serum VEGF levels after 2 hours (Fig 2a). As expected, the B16-F10 cell-induced accumulation of the Evans blue tracer was anatomically evident in lungs (Fig 2b) and accounted for a 2.5-fold increase in vascular permeability relative to PBS treated mice (Fig 2c). VIAN-c4551 prevented the upregulation of lung vascular permeability in response to melanoma cells without affecting basal vasopermeability in the absence of cells (Fig 2b and c). Because VEGF is a major vascular permeability stimulator produced by tumor cells, including B16-F10 cells [2,3], and VIAN-c4551 inhibits VEGF-induced vasopermeability [19], the circulating levels of VEGF were determined 2 hours post-tumor cell inoculation or PBS injection (Fig 2a). B16-F10 cells increased serum VEGF levels, and VEGF values were not altered by VIAN-c4551 (Fig 2d). We conclude that VIAN-c4551 abrogates lung vascular permeability not by blocking the melanoma cell release of VEGF but by inhibiting the vasopermeability action of the VEGF released by melanoma cells.

Because a single i.v. injection of VIAN-c4551 prevented melanoma cell-induced pulmonary leakage at 2 hours, when melanoma cells are circulating [26], we hypothesized that VIAN-c4551 decreases the vascular exit of tumor cells and, thereby, reduces lung melanoma metastasis.

## A single administration of VIAN-c4551 reduces melanoma metastatic nodules in lungs

To test whether the inhibition of lung vascular leakage by VIAN-c4551 interferes with pulmonary metastasis, VIAN-c4551 or vehicle (saline) was injected i.v. 30 minutes before the systemic inoculation of B16-F10 cells and lungs were collected after 17 days to evaluate metastatic nodules (Fig 3a). Representative images showing the ventral and dorsal views of left and right pulmonary lobes showed many superficial macroscopic black nodules in vehicle-treated mice and only a few in mice treated with VIAN-c4551 (Fig 3b). In agreement, VIAN-c4551 reduced by 55% the number and by 50% the size of macroscopic nodules at the lung surface (Fig 3c and d). Moreover, the histological evaluation of five longitudinal

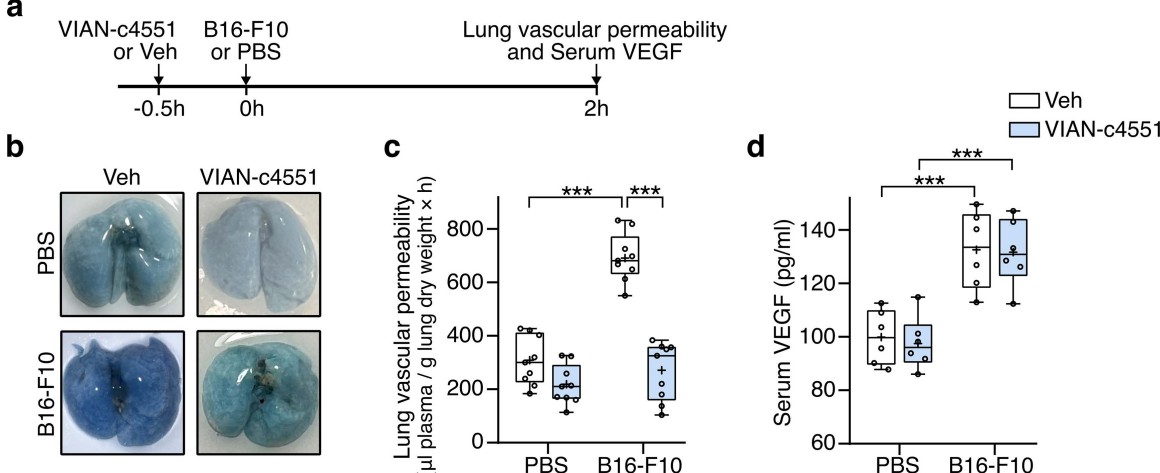

**Fig 2. VIAN-c4551 prevents the increase of vascular permeability stimulated by melanoma cells. (a)** Timeline of the experiment: VIAN-c4551 or vehicle (Veh) were i.v. injected 30 minutes before the i.v. delivery of B16-F10 cells or PBS. Pulmonary vascular permeability or VEGF levels were evaluated 2 hours post-tumor cells or PBS. **(b)** Photographs of representative lungs showing the accumulation of Evans blue-stained albumin as index of pulmonary vascular permeability. **(c)** Quantification of lung vascular permeability in Veh- and VIAN-c4551-treated mice after i.v. injection of PBS or B16-F10 cell inoculation. Data from 3 independent experiments (n = 9) are graphed in a box and a whisker plot; the box frames the interquartile range, the horizontal line indicates the median, and the whiskers the min and max values. The mean is indicated by +. ***P < 0.001 (Two-way ANOVA, Sidak's multiple comparison test). **(d)** VEGF levels in serum from Veh- and VIAN-c4551-treated mice after injection of PBS or B16-F10 cell inoculation. Values are box and whiskers plot from 2 independent experiments (n = 6), ***P < 0.001 (Two-way ANOVA, Sidak's multiple comparison test).

lung sections showed that VIAN-c4551 reduced by 56% and by 57% the number and size of microscopic melanoma nodules, respectively (Fig 3e-g). These findings implied that the inhibition of melanoma cell-induced vascular leakage by VIAN-c4551 is sufficient to strongly inhibit B16-F10 cell metastasis in lungs and prompted addressing this mechanism further.

### VIAN-c4551 inhibits the melanoma cell-induced permeability of endothelial cell monolayers mediated by VEGF

Transendothelial electrical resistance (TEER) of cellular monolayers measures the integrity of cell junctions and it is a quantitative indicator of the monolayer permeability [27]. B16-F10 cells (Fig 4a) and their conditioned media (B16-F10-CM) (Fig 4b) decreased the TEER of bovine umbilical vein endothelial cell (BUVEC-E6E7) monolayers throughout a 6-hour incubation period. VIAN-c4551 prevented the reduction of TEER induced by both B16-F10 and B16-F10-CM while VIAN-c4551 alone had no effect. Likewise, the anti-VEGF monoclonal fragment ranibizumab blocked the TEER reduction in response to B16-F10 and B16-F10-CM, indicating that VEGF is a substantial contributor to melanoma cell-induced hyperpermeability (Fig 4a and b). In agreement, VEGF accumulated overtime in the B16-F10-CM (Fig 4c) and, as expected from in vivo data (Fig 2d), its levels were not affected by VIAN-c4551. Increased vascular permeability due to weakened endothelial cell junctions associates with the redistribution of the actin cytoskeleton that leads to cell contraction [28,29]. The actin cytoskeleton was therefore evaluated by phalloidin-TRITC staining of BUVEC-E6E7 monolayers exposed 1 hour to the B16-F10-CM. The B16-F10-CM altered the actin cytoskeleton as revealed by increased fluorescence and frequent contracted cell bodies and these changes were prevented by VIAN-c4551 and anti-VEGF (Fig 4d). To evaluate the functional link between vasopermeability and metastatic activity, we turned to 3T3 cells, a mouse cell line derived from embryonic fibroblasts that is devoid of metastatic activity [30]. Their conditioned media (3T3-CM) did not modify the TEER of BUVEC-E6E7 monolayers throughout a 6-hour incubation period (Fig 4e). BUVEC-E6E7 is a well characterized cell line that maintains the endothelial cell phenotype, including the response to specific regulators (such as VEGF), throughout extended replication times without signs of senescence and genetic instability [20]. We confirmed

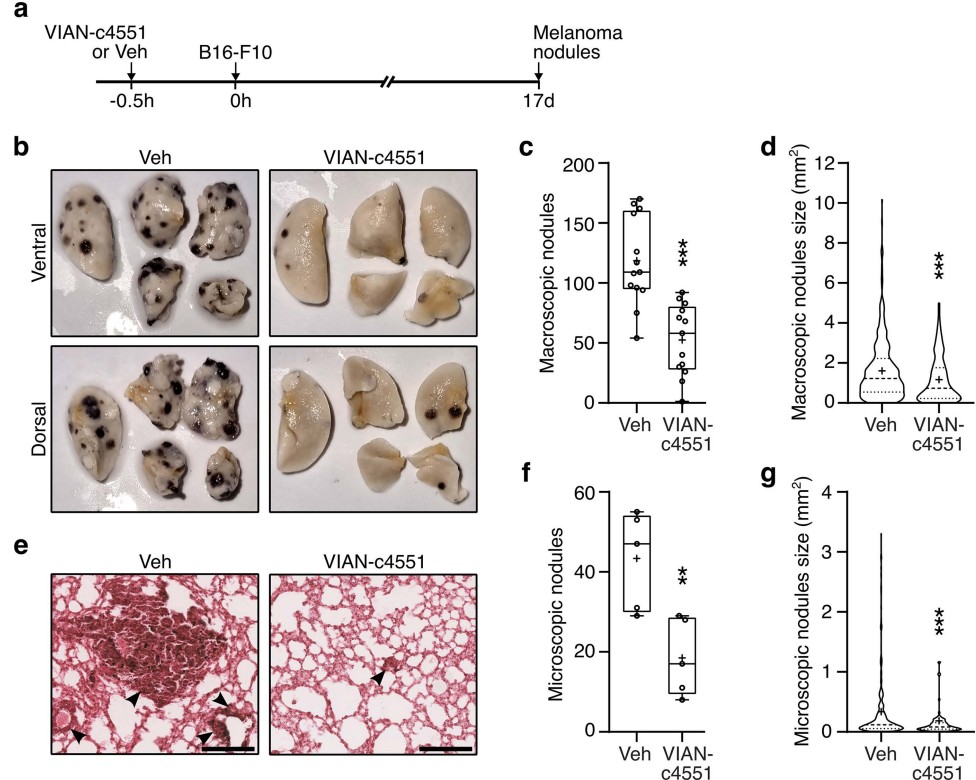

**Fig 3. A single administration of VIAN-c4551 reduces the number and size of lung melanoma metastases in mice. (a)** Timeline of the experiment: VIAN-c4551 or vehicle (Veh) were injected i.v. 30 minutes before the i.v. inoculation of B16-F10 cells. Lung melanoma nodules were evaluated 17 days post-tumor cell delivery. **(b)** Representative ventral and dorsal views of the left lung and the four right lobes of a mice treated with Veh or VIAN-c4551. Quantification of the number **(c)** and size **(d)** of macroscopic melanoma nodules on the lung surface. **(e)** Representative lung sections stained with hematoxylin/eosin showing microscopic melanoma nodules (arrows) (scale = 200 µm). Quantification of the number **(f)** and size **(g)** of internal microscopic melanoma nodules in lungs. Numbers of nodules are graphed in a box and a whisker plot; the box represents the interquartile range, the horizontal line indicates the median, and the whiskers the min and max values. Size is graphed in a violin plot showing median and quartiles. The mean is indicated by +. Data from 3 independent experiments (n = 13). **P = 0.008, ***P < 0.001 vs Veh (unpaired t-test).

the adequacy of BUVEC-E6E7 by obtaining similar findings in primary cultures of human umbilical vein endothelial cells (HUVEC), the most frequently used in vitro model in oncology or cardiovascular endothelial cell research [31]. Like in BUVEC-E6E7, the CM from B16-F10 melanoma cells decreased the TEER of HUVEC monolayers and such reduction was prevented by VIAN-c4551, whereas VIAN-c4551 alone had no effect (Fig 4f). Furthermore, VIAN-c4551 prevented the decrease of TEER induced by the CM from a highly metastatic human breast cancer cell line (MDA-MB-231) [32], while the CM from a low-metastatic breast cancer cell line (MCF-7) [32,33] did not modify the TEER in HUVEC.

These findings supported the inhibition by VIAN-c4551 of VEGF-mediated endothelial cell permeability in response to melanoma cells and other metastatic cancer cells and encouraged investigating whether this action interfered with the ability of B16-F10 cells to extravasate.

## VIAN-c4551 inhibits the transendothelial migration of B16-F10 cells in vitro and in vivo

The effect of VIAN-c4551 on the extravasation of melanoma cells was evaluated in vitro by measuring the migration of melanoma cells across a tight BUVEC-E6E7 monolayer. In this assay, fluorescent B16-F10-GFP cells were seeded at the top (luminal side) of a BUVEC-E6E7 confluent monolayer and the number of fluorescent cells found at the bottom

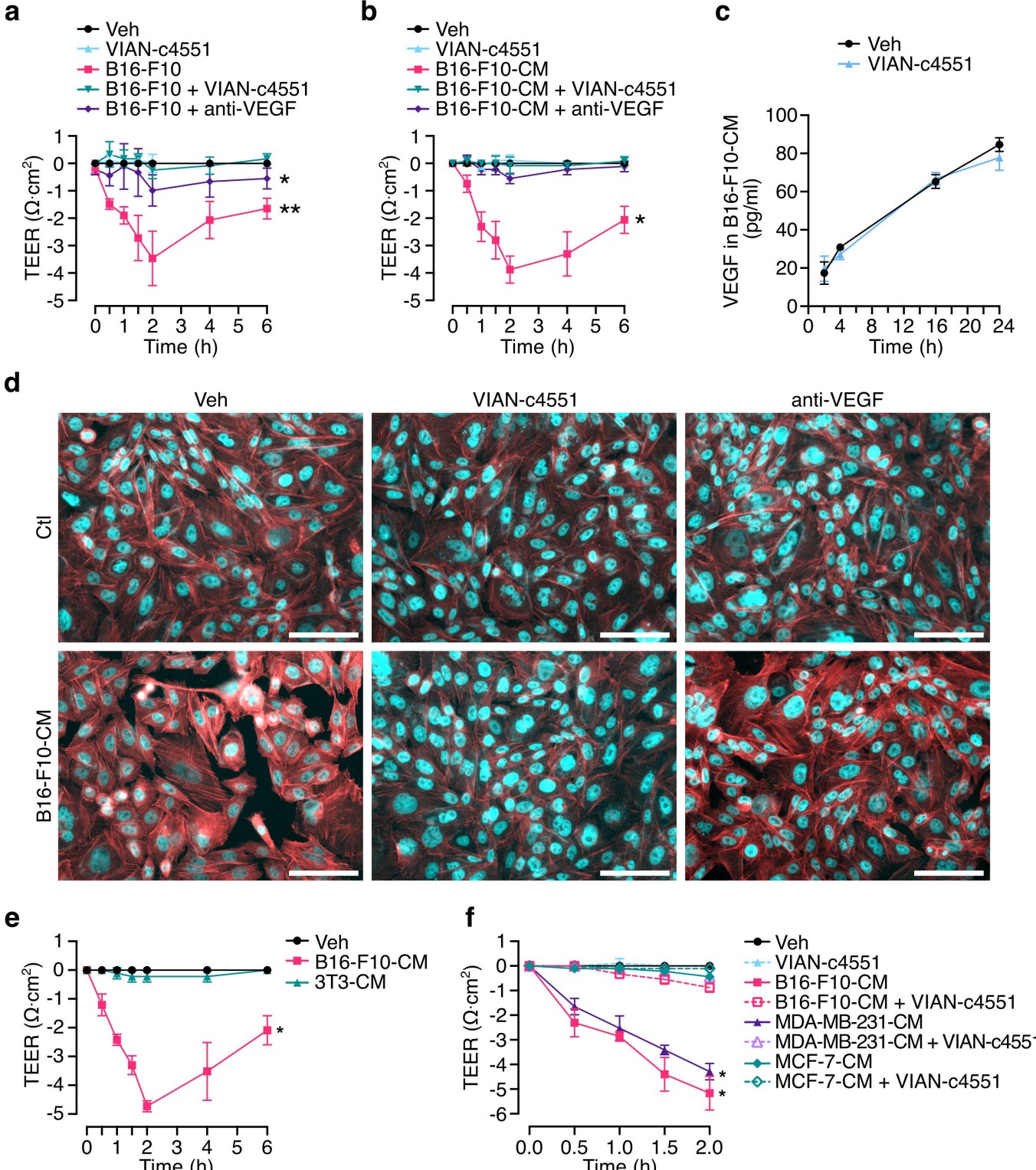

**Fig 4. VIAN-c4551 inhibits the melanoma cell-induced permeability of endothelial cell monolayers mediated by VEGF. (a)** Effect of B16-F10 cells on the transendothelial electrical resistance (TEER) of bovine umbilical vein endothelial cell line (BUVEC-E6E7) monolayers in the absence or presence of VIAN-c4551 or anti-VEGF over a 6-hour period. *P = 0.0168, **P = 0.0081 vs Veh. **(b)** Effect of B16-F10 conditioned media (B16-F10-CM)

on TEER of BUVEC-E6E7 monolayers. *P = 0.0182 vs Veh. (Repeated measurements one-way ANOVA, Dunnett's multiple comparisons test). Values are means ± SD of 3 independent experiments. **(c)** VEGF levels in the conditioned medium of B16-F10 melanoma cells treated or not with VIAN-c4551 throughout a 24-hour incubation period. Values are means ± SD of 3 independent experiments. **(d)** Representative images of the actin cytoskeleton distribution in BUVEC-E6E7 monolayers treated or not with B16-F10-CM in the absence or presence of VIAN-c4551 or anti-VEGF (scale = 100 μm). **(e)** Effect of 3T3 conditioned media (3T3-CM) and B16-F10 conditioned media (B16-F10-CM) on TEER of BUVEC-E6E7 monolayers. *P = 0.0105 vs Veh. (Repeated measurements one-way ANOVA, Dunnett's multiple comparisons test). **(f)** Effect of B16-F10 (B16-F10-CM), MDA-MB-231 (MDA-MB-231-CM), and MCF-7 (MCF-7-CM) conditioned media on TEER of human umbilical vein endothelial cell (HUVEC) monolayers. *P < 0.001 vs Veh. (Two-way ANOVA, Dunnett's multiple comparisons test). Values are means ± SD of 3 independent experiments.

(abluminal side) of the monolayer indicated transendothelial tumor cell migration. VIAN-c4551 reduced by 54% the number of B16-F10-GFP cells crossing the endothelial cell monolayer (Fig 5a and b), thereby supporting the inhibition of tumor cell extravasation by VIAN-c4551. Extravasation of melanoma cells was also studied in vivo. Mice were injected i.v. with VIAN-c4551 or vehicle (saline) 30 minutes before the i.v. inoculation of B16-F10-GFP and 2 hours later, mice were perfused and lungs collected to visualize tumor cell extravasation in the premetastatic lung through the histological presence of fluorescent tumor cells (Fig 5c) or the mRNA expression levels of GFP (Fig 5d). VIAN-c4551-treated mice demonstrated lower numbers of fluorescent tumor cells and reduced lung expression of GFP relative to controls.

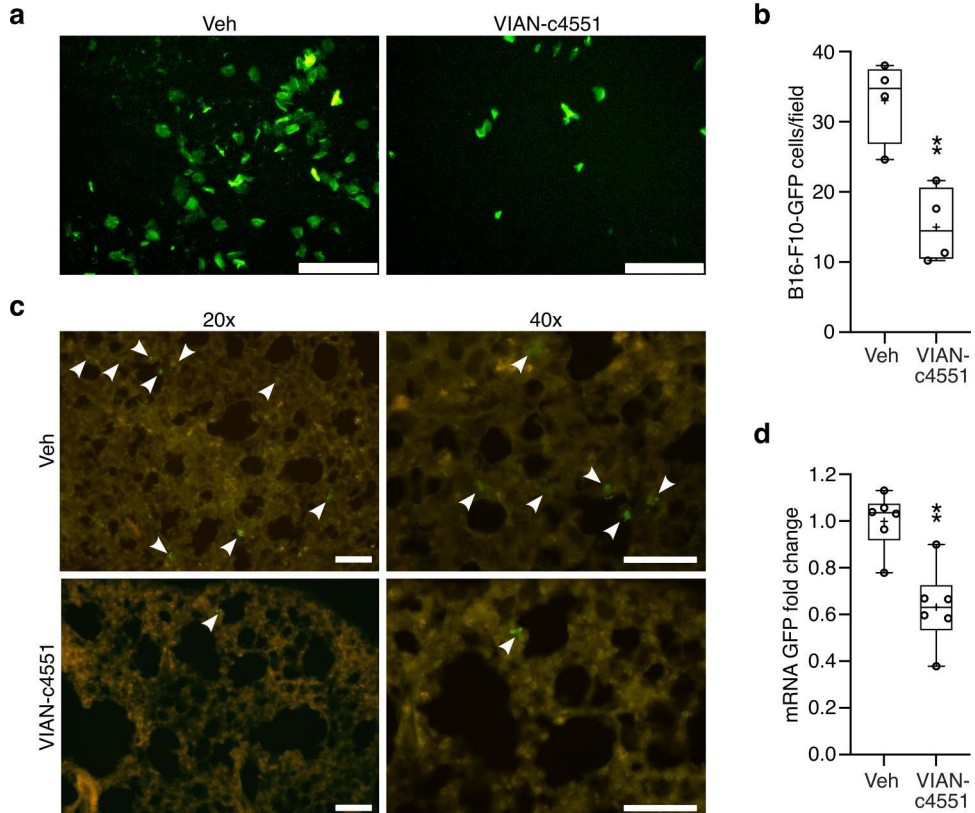

**Fig 5. VIAN-c4551 reduces the in vitro transendothelial migration and the lung extravasation of B16-F10 melanoma cells. (a)** Representative images showing the B16-F10 cells expressing GFP that migrated across a confluent endothelial cell monolayer in the presence of vehicle (Veh) or VIAN-c4551. **(b)** Quantification of the number of migrated B16-F10-GFP cells. Values are box and whiskers plot from 4 independent experiments. The mean is indicated by +. **P = 0.0042 vs Veh (unpaired t-test) (scale = 120 μm). **(c)** Representative lung sections showing extravasated B16-F10-GFP cells 2 hours after their inoculation (arrows) (scale = 100 μm). **(d)** GFP mRNA levels in lungs from Veh- and VIAN-c4551-treated mice. Values are box and whiskers plot from 2 independent experiments (n = 6). The mean is indicated by +. **P = 0.0014 vs Veh (unpaired t-test).

## Subacute toxicity evaluation of VIAN-c4551

Due to its therapeutic potential, we performed a sub-acute toxicity study in mice daily injected i.v. with 2 doses of VIAN-c4551 for 14 days. No significant changes were observed in body weight, biochemical, and hematological parameters at either dose (Tables 1 and 2). Likewise, the weight of body organs such as the heart, lungs, liver, spleen and kidneys were similar among groups, and the histopathological examination of these organs revealed regular parenchyma architecture (data not shown).

Data are expressed as mean (SD), n = 6. One way ANOVA. HCT, hematocrit; Hb, hemoglobin; RBC, red blood cells; MCV, mean corpuscular volume; MCHC, mean corpuscular hemoglobin concentration; WBC, white blood cells; NEUT, neutrophils; LYM, lymphocytes; MON, monocytes. Reference values were from Charles River Technical Information Sheet for C57BL6/J mice, except for MON [36].

## Discussion

Metastasis is divided into pre-metastatic and post-metastatic phases that depend on whether tumor cells have or not engrafted at the metastatic region [2]. Because metastasis from a primary tumor can occur before primary cancer is detected [37], there is an urgent need to understand and target the premetastatic process involving the local invasion

**Table 1. Body weight and serum biochemical parameters of mice treated with VIAN-c4551 i.v. for 14 days. Data are expressed as mean (SD), n = 6. One way ANOVA. i.v., intravenous; BW, body weight; Cr, creatinine; ALT, alanine aminotransferase; AST, aspartate aminotransferase; Glu, glucose; ALB, albumin; TP, total protein. Reference values were from Charles River Technical Information Sheet for C57BL6/J mice [34], except for Urea [35] and Glu [36].**

| Parameters | Reference values | Vehicle | VIAN-c4551 1 mg/kg/d | VIAN-c4551 10 mg/kg/d | P |
|---|---|---|---|---|---|
| BW (g) | 16.70 - 20.10 | 16.59 (1.04) | 17.47 (1.02) | 16.97 (0.89) | 0.50 |
| Cr (mg/dl) | 0.20 - 0.50 | 0.36 (0.02) | 0.35 (0.02) | 0.34 (0.02) | 0.15 |
| Urea (mmol/l) | 9.27 - 12.09 | 9.75 (0.78) | 10.14 (0.99) | 9.34 (0.98) | 0.26 |
| ALT (IU/l) | 27 - 195 | 67.17 (31.19) | 57.60 (35.44) | 84.33 (73.19) | 0.29 |
| AST (IU/l) | 43 - 397 | 228 (55.57) | 224.6 (110.23) | 250.83 (54.1) | 0.80 |
| Glu (mmol/l) | 4.59 - 9.89 | 6.20 (2.65) | 7.49 (1.24) | 6.30 (1.03) | 0.58 |
| ALB (g/dl) | 2.40 - 4.30 | 3.30 (0.24) | 3.42 (0.16) | 3.46 (0.25) | 0.16 |
| TP (g/dl) | 4.80 - 7.20 | 5.93 (0.72) | 5.90 (0.21) | 5.93 (0.46) | 0.98 |

**Table 2. Hematological parameters of mice treated with VIAN-c4551 i.v. for 14 days. Data are expressed as mean (SD), n = 6. One way ANOVA. HCT, hematocrit; Hb, hemoglobin; RBC, red blood cells; MCV, mean corpuscular volume; MCHC, mean corpuscular hemoglobin concentration; WBC, white blood cells; NEUT, neutrophils; LYM, lymphocytes; MON, monocytes. Reference values were from Charles River Technical Information Sheet for C57BL6/J mice [34], except for MON [36].**

| Parameters | Reference values | Vehicle | VIAN-c4551 1 mg/kg/d | VIAN-c4551 10 mg/kg/d | P |
|---|---|---|---|---|---|
| HCT (%) | 37.20 - 58.00 | 47 (4) | 49 (4) | 48 (2) | 0.85 |
| Hb (g/dl) | 10.90 - 18.10 | 14.90 (1.19) | 15.31 (1.17) | 15.11 (0.57) | 0.78 |
| RBC (x10$^{12}$/l) | 7.37 - 11.50 | 10.35 (0.94) | 10.65 (1.01) | 10.47 (0.45) | 0.90 |
| MCV (fl) | 42.60 - 55.60 | 45.73 (0.99) | 45.70 (1.12) | 45.58 (0.67) | 0.89 |
| MCHC (g/dl) | 26.00 - 35.90 | 31.35 (0.32) | 31.38 (3.19) | 31.58 (5.60) | 0.74 |
| WBC (x10$^9$/l) | 3.90 - 13.94 | 6.63 (1.56) | 6.77 (2.78) | 5.57 (1.20) | 0.37 |
| NEUT (x10$^9$/l) | 0.42 - 2.55 | 2.12 (1.67) | 1.40 (0.98) | 1.07 (0.86) | 0.32 |
| LYM (x10$^9$/l) | 2.88 - 10.92 | 4.45 (1.98) | 5.32 (2.59) | 4.48 (1.56) | 0.74 |
| MON (x10$^9$/l) | 0.02 - 0.04 | 0.05 (0.08) | 0.03 (0.05) | 0.07 (0.08) | 0.61 |

and intravasation of tumor cells at the primary site and their extravasation in remote organs [1,2]. Notably, primary tumors stimulate extravasation by increasing the permeability of blood vessels at metastatic sites [2,3,25,38,39]. Here, we support the inhibition of distant vascular permeability for the prevention of tumor metastasis and unveil the anti-vasopermeability factor VIAN-c4551 as a potential therapeutic drug able to prevent metastasis generation.

Melanoma (B16-F10) and breast (MDA-MB-231) cancer cell lines increase the permeability of lung blood vessels before metastasis by the production of vasoactive factors such as angiopoietin (Angpt) 2 and 4, matrix metalloproteinases (MMP) 1, 2, 3, and 10 and downstream effectors of VEGF (epiregulin and cyclooxygenase 2). These factors increase the permeability of lung capillaries that facilitates the transendothelial passage of tumor cells to seed pulmonary metastases [25,38,39]. Because increased vascular permeability favors extravasation, an essential step in pre-metastatic progression, inhibitors of vasopermeability represent promising therapeutics. One such inhibitor is VIAN-c4551, the cyclic heptapeptide analog of the endogenous anti-angiogenic protein vasoinhibin.

Vasoinhibin inhibits the increase in vascular permeability induced by VEGF by promoting the dephosphorylation/inactivation of endothelial nitric oxide synthase (eNOS) leading to the inhibition of the production of nitric oxide [40], an important signal that destabilizes the endothelial cell barrier through posttranslational modifications of adherent junction proteins [41]. VIAN-c4551 also inhibits the VEGF-induced permeability of endothelial cell monolayers [19] and both VIAN-c4551 [19] and vasoinhibin [40,42] protect against disruption of the blood retinal barrier in experimental models of diabetic retinopathy. However, beneficial effects against the extravasation of premetastatic tumors remain to be determined.

In our study, we inoculated B16-F10 cells into the tail vein of mice to circumvent primary tumor formation with the advantage of establishing a correlation between short-term lung hypervasopermeability and rapid metastatic progression. We showed that melanoma cells maximally increase pulmonary vascular permeability and extravasate at 2 hours post inoculation and that hypervasopermeability lasted at least 8 hours. The timing is consistent with the reported 8 hours required for melanoma cells to extravasate after their systemic delivery [26] and, thereby, with their metastatic seeding. Furthermore, enhanced vasopermeability at 2 hours associated with the upregulation of circulating VEGF levels suggesting the release of VEGF by melanoma cells as responsible mechanism. Accordingly, we hypothesized that the acute treatment with a VEGF inhibitor such as VIAN-c4551 would block the increase in lung vascular permeability and interfere with the extravasation process leading to fewer pulmonary metastasis. In agreement, a single intravascular injection of VIAN-c4551 inhibited the early increase in lung vasopermeability and the extravasation of melanoma cells and was enough to substantially reduce the number and size of macroscopic and microscopic metastatic nodules in lungs.

Inhibition of lung vascular leakage was previously reported to limit the number of B16-F10 cells exiting the circulation and their metastatic spread [25]. However, in contrast to our study, vasopermeability was evaluated in the presence of the primary melanoma tumor and after longer times (days) following B16-F10 cell intravascular inoculation. The authors proposed as a mechanism the altered lung microenvironment by the primary tumor through the local expression of vasopermeability factors, namely Angpt 2, MMP3, and MMP10, but not VEGF [25]. However, VEGF circulating levels were not evaluated.

The contribution of systemic VEGF to tumor cell extravasation is consistent with circulating cancer cells producing VEGF that is directly related to their metastatic potential [43] and with highly metastatic cancer cells circulating as multicellular clusters [44] whose extravasation is favored by vascular leakage. Also, the VEGF inhibitor VIAN-c4551 blocked melanoma cell-induced lung vasopermeability, melanoma cell extravasation, and metastatic growth. By comparing the vasopermeability effect on TEER values of B16-F10 cells to that of non-metastatic cells (3T3 cells) or breast cancer cells with low (MCF7) and high (MDA-MB-231) metastatic potential, our in vitro data supported the functional link between increased vasopermeability and metastatic activity and extended the putative anti-metastatic influence of VIAN-c4551 to other types of cancer. Moreover, our in vitro findings showed that VEGF secreted by B16-F10 cells perturbs endothelial cell junctions in a manner that facilitates the passage of tumor cells across endothelial cell monolayers and that VIAN-c4551 prevented melanoma cell-induced vascular leakage like the anti-VEGF antibody fragment ranibizumab,

which blocks the interaction of VEGF with its receptor [45]. Of note, VIAN-c4551 did not reduce the upregulation of VEGF circulating levels following melanoma cell inoculation nor the release of VEGF by cultured melanoma cells indicating that VIAN-c4551 targets the action and not the production of VEGF.

Our study does not rule out that other vasopermeability factors produced by melanoma cells are targeted by VIAN-c4551. Vasoinhibin is a broad acting agent that inhibits the action of different vasoactive factors [5]. It binds to a multi-protein complex that includes plasminogen activator inhibitor 1 (PAI-1), urokinase, and the urokinase receptor [46] in endothelial cell membranes which can contribute to the inhibition of multiple signaling pathways (Ras-Raf-MAPK, Ras-Tiam1-Rac1-Pak1, PI3K-Akt-eNOS, and PLCγ-eNOS) activated by several proangiogenic and vasopermeability factors (VEGF, basic fibroblast growth factor, bradykinin, and interleukin-1β) [5]. It is unclear how binding to the multi-protein complex inhibits endothelial cells angiogenic processes [46], and the contribution of other vasoinhibin-binding proteins [47] and/or interacting molecules [48] likely participate. Along this line, VIAN-c4551 does not bind to PAI-1 [49], yet it inhibits VEGF-induced vasopermeability [19; present results] and blood vessel growth [4].

Vasoinhibin reduces vascular leakage by blocking the phosphorylation/activation and the calcium-calmodulin binding/activation of eNOS in response to VEGF, bradykinin, acetylcholine, diabetic vitreous, and arthritic joints [40,42,50,51]. Likewise, vasoinhibin inhibits the action of several proangiogenic factors (VEGF, bFGF, bradykinin, IL-1β) with an impact on angiogenesis-dependent diseases, including cancer [5,12]. In fact, there is evidence that vasoinhibin inhibits the growth and vascularization of melanoma metastatic nodules in lungs during the post-metastatic phase [16], i.e., after pulmonary engraftment and colonization [1,26]. In that study [16], an adenoviral vector encoding the vasoinhibin isoform of 16 kDa (16k PRL) delivered 2 days post-melanoma cell intravascular inoculation, reduced the number and size of lung metastases. The tardy 2-day treatment and the delayed upregulation of vasoinhibin by gene therapy precluded evaluating the influence of vasoinhibin at the pre-metastatic phase.

A major obstacle for addressing vasoinhibin action in animal models has been the lack of sufficient functional protein. Post-translational modifications, protein folding, and instability complicate the recombinant production of vasoinhibin [18] and the delivery of sufficient quantities of vasoinhibin needed the gene therapy approach [16,52]. These difficulties have been overcome by the development of VIAN-c4551, a potent, easy-to-produce vasoinhibin analog [4]. VIAN-c4551 uncovered inhibition of vascular leakage as a novel mechanism by which vasoinhibin could block cancer cell extravasation at the pre-metastatic phase. However, the effect may be stronger in VIAN-c4551 because it lacks the ability of vasoinhibin to upregulate adhesion molecules in endothelial cells [49,53] that favor melanoma-endothelial cell interaction for extravasation [54]. Furthermore, VIAN-c4551 may be regarded as safe as it did not cause any significant change in hematological, biochemical, and histological parameters in our sub-acute toxicity study.

In conclusion, our in vitro and in vivo studies support vascular leakage as a limiting step for cancer cell extravasation and disclose VIAN-c4551 as a potential anti-cancer agent for the early prevention of metastatic spread that warrants further research.

## Acknowledgments

The authors thank Fernando López Barrera, Xarubet Ruíz Herrera, Adriana González Gallardo, Ericka A. de los Ríos Arellano, Nydia Hernández Ríos, Moisés Mendoza Baltazar, Alejandra Castilla León, María A. Carbajo Mata, Eugenia Ramos Aguilar, and Martín García Servín for their excellent technical assistance.

## Author contributions

**Conceptualization:** Alma Lorena Perez, Juan Pablo Robles, Carmen Clapp.

**Data curation:** Alma Lorena Perez.

**Formal analysis:** Alma Lorena Perez, Juan Pablo Robles, Carmen Clapp.

**Funding acquisition:** Carmen Clapp.

**Investigation:** Alma Lorena Perez, Magdalena Zamora, Manuel Bahena, Regina Aramburo-Williams, Elva Adan-Castro, Daniela Granados-Carrasco, Juan Pablo M. Robles.

**Methodology:** Alma Lorena Perez, Magdalena Zamora, Elva Adan-Castro, Juan Pablo Robles.

**Project administration:** Juan Pablo Robles, Carmen Clapp.

**Resources:** Thomas Bertsch, Jakob Triebel, Carmen Clapp.

**Supervision:** Gonzalo Martinez de la Escalera, Juan Pablo Robles, Carmen Clapp.

**Writing – original draft:** Alma Lorena Perez, Carmen Clapp.

**Writing – review & editing:** Alma Lorena Perez, Magdalena Zamora, Elva Adan-Castro, Thomas Bertsch, Jakob Triebel, Gonzalo Martinez de la Escalera, Juan Pablo Robles, Carmen Clapp.

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
