## [Decision Letter · Decision Letter 0]

31 Jan 2025

PONE-D-24-58885The antiangiogenic peptide VIAN-c4551 inhibits lung melanoma metastasis in mice by reducing pulmonary vascular permeabilityPLOS ONE

Dear Dr. Clapp,

Thank you for submitting your manuscript to PLOS ONE. After careful consideration, we feel that it has merit but does not fully meet PLOS ONE’s publication criteria as it currently stands. Therefore, we invite you to submit a revised version of the manuscript that addresses the points raised during the review process.

We look forward to receiving your revised manuscript.

Kind regards,

Mohamed Abdelkarim

Academic Editor

PLOS ONE

“This work was supported by the Dirección General de Asuntos del Personal Académico (DGAPA) Universidad Nacional Autónoma de México (UNAM) (Grant IN202424) to CC. Alma Lorena Perez is a doctoral student from the ‘Programa de Doctorado en Ciencias Biomédicas, Universidad Nacional Autónoma de México (UNAM)’ and has received fellowship 788879 from Consejo Nacional de Humanidades, Ciencia y Tecnología (CONAHCYT). MZ, EA-C and JPR are postdoctoral fellows from CONAHCYT.”

3. We note that you have a patent relating to material pertinent to this article. Please provide an amended statement of Competing Interests to declare this patent (with details including name and number), along with any other relevant declarations relating to employment, consultancy, patents, products in development or modified products etc. Please confirm that this does not alter your adherence to all PLOS ONE policies on sharing data and materials, as detailed online in our guide for authors http://journals.plos.org/plosone/s/competing-interests by including the following statement: "This does not alter our adherence to  PLOS ONE policies on sharing data and materials.” If there are restrictions on sharing of data and/or materials, please state these. Please note that we cannot proceed with consideration of your article until this information has been declared.

Reviewers' comments:

Reviewer's Responses to Questions

**Comments to the Author**

1. Is the manuscript technically sound, and do the data support the conclusions?

Reviewer #1: Yes

Reviewer #2: Yes

2. Has the statistical analysis been performed appropriately and rigorously? 

Reviewer #1: Yes

Reviewer #2: I Don't Know

3. Have the authors made all data underlying the findings in their manuscript fully available?

Reviewer #1: Yes

Reviewer #2: Yes

4. Is the manuscript presented in an intelligible fashion and written in standard English?

Reviewer #1: Yes

Reviewer #2: Yes

5. Review Comments to the Author

Reviewer #1: In this manuscript by Perez et al, the authors investigated the function of VIAN-c4551, a peptide analogue of vasoinhibin, in extravasation and lung metastasis of melanoma cells. They discovered that VIAN-c4551 inhibits B16-F10 cell-induced increase in vascular permeability and metastasis in vivo, as well as endothelial monolayer permeability in vitro. This is a well-designed study. However, there are a number of concerns that need to be addressed before this manuscript is in a publishable fashion. Specific comments are as follows:

1) In the Evans blue assay, the quantified data were shown as the dye concentration in the extracts (as described in the Methods). The unit is ul/ g lung x h. What the "ul" represents needs to be defined.

2) Is there a particular reason that female animals were used in this study?

3) Although TEER and transendothelial migration assays are established methods. It is advised to put in more details on for example, how to define BUVEC-E6E7 monolayers.

4) Do non-metastatic cancer cells (and their conditioned medium) affect TEER as there is no control for this part?

5) In the discussion, in Line 336, the authors stated that VIAN-c4551 inhibits the increase in vascular permeability induced by VEGF by blocking production of NO by eNOS. Is it shown in ref. 19? Please rephrase if it is not a published result.

6) The results show that VIAN-c4551 blocks metastatic cell-induced increase in vascular permeability, presumably through suppressing VEGF action without affecting VEGF expression. So how does this work? As this manuscript does not have data for mechanistic investigation, the authors should at least discuss potential molecular mechanisms such as putative receptors, signaling pathways, distinct features from vasoinhibin etc.

7) Minor: a number of typos: line 90 - SFB; line 158 - lung paraffin sections from lungs

Reviewer #2: The study titled “The antiangiogenic peptide VIAN-c4551 inhibits lung melanoma metastasis in mice by reducing pulmonary vascular permeability” demonstrates that VIAN-c4551 effectively inhibits lung melanoma metastasis in mice by reducing pulmonary vascular permeability and preventing melanoma cell extravasation. The findings are well-supported by both in vivo and in vitro experiments, and the study opens new avenues for the development of anti-metastatic therapies. However, further research is needed to validate these findings in human clinical trials and to explore the full range of mechanisms by which VIAN-c4551 exerts its anti-metastatic effects.

To further evaluate the study’s findings and address potential limitations, the following questions could be posed to the authors:

Dose Selection and Optimization:

- Why was a single dose of VIAN-c4551 (1 mg/kg) chosen for the experiments? Have you conducted dose-response studies to determine the optimal therapeutic dose?

- Have you explored the pharmacokinetics of VIAN-c4551, such as its half-life, bioavailability, or potential accumulation in tissues, which could influence its efficacy and safety?

Toxicity and Safety:

- Did you evaluate the potential toxicity of VIAN-c4551 in mice, particularly with long-term administration? Were there any signs of systemic toxicity, such as organ damage or immune suppression?

- Why did you not assess other serum markers (e.g., liver enzymes, kidney function) to evaluate the systemic impact of VIAN-c4551 treatment?

In Vitro Model Selection:

- Why did you choose bovine umbilical vein endothelial cells (BUVEC-E6E7) for the in vitro experiments? Did you consider using human endothelial cell lines to better reflect human physiology?

- Why did you focus on venous endothelial cells rather than arterial endothelial cells, which might be more relevant to the pulmonary vasculature and metastatic seeding?

6. PLOS authors have the option to publish the peer review history of their article (what does this mean? ). If published, this will include your full peer review and any attached files.

**Do you want your identity to be public for this peer review?** For information about this choice, including consent withdrawal, please see our Privacy Policy .

Reviewer #1: No

Reviewer #2: No

---

## [Author Response · Author response to Decision Letter 0]

4 Apr 2025

Reviewer 1:

In this manuscript by Perez et al, the authors investigated the function of VIAN-c4551, a peptide analogue of vasoinhibin, in extravasation and lung metastasis of melanoma cells. They discovered that VIAN-c4551 inhibits B16-F10 cell-induced increase in vascular permeability and metastasis in vivo, as well as endothelial monolayer permeability in vitro. This is a well-designed study.

However, there are a number of concerns that need to be addressed before this manuscript is in a publishable fashion.

Reply: We thank the reviewer for the kind comments and careful evaluation of our work.

Specific comments are as follows:

1) In the Evans blue assay, the quantified data were shown as the dye concentration in the extracts (as described in the Methods). The unit is ul/ g lung x h. What the "ul" represents needs to be defined.

Reply: µl indicates the volume of extravasated plasma in lungs. We have clarified this information by modifying the legend of the Y axis in Figures 1c and 2c to “µl plasma /g lung dry weight x h”.

2) Is there a particular reason that female animals were used in this study?

Reply: The preceding work showing the inhibition of melanoma tumor growth and neovascularization by vasoinhibin (Nguyen et al., 2007) and VIAN-c4551 (Robles et al., 2022) was performed in female adult C57/BL6 mice. Because the purpose of the current study was to investigate whether the anti-tumor action of VIAN-c4551 extended to the inhibition of melanoma cell extravasation, we continued using the female as working model. However, there are sex differences in melanoma tumor progression that may relate to a more robust immune system in females (Dakup et al., 2020). Addressing the anti-tumor action of VIAN-c4551 on both sexes warrants further research. The use of females has been explained in the revised text (lines 134-136).

3) Although TEER and transendothelial migration assays are established methods. It is advised to put in more details on for example, how to define BUVEC-E6E7 monolayers.

Reply: Descriptions were extended in the revised manuscript as follows:

Vascular permeability in vitro (lines 94-98):

BUVEC-E6E7 were seeded on 6.5 mm transwell clear polyester membrane inserts (Corning) with pore sizes of 0.4 μm at an initial density of 7.2 x 103 cells per well (30% confluency). After approximately 96 hours, transendothelial electrical resistance (TEER) measured by the EVOM2 Epithelial Voltohmmeter (World Precision Instruments, Sarasota, FL) stabilized around 55 Ω.cm2 (reflecting a confluent monolayer) and treatments started.

Transendothelial migration assay (lines 124-128):

BUVEC-E6E7 (7.2 x 103) were seeded on 6.5 mm transwell membrane inserts with 8 μm pores coated with 0.38 mg/ml matrigel (354234, Corning). Upon confluency (55 Ω.cm2, revealed by the EVOM2 resistance tester) monolayers were treated for 1 hour with 100 nM VIAN-c4551 into the upper (luminal) chamber followed by the addition of fluorescent B16-F10-GFP cells (3.5 x 104 cells per well).

4) Do non-metastatic cancer cells (and their conditioned medium) affect TEER as there is no control for this part?

Reply: Attending the reviewer’s concern, we now show that the NIH/3T3 mouse cell line derived from embryonic fibroblasts devoid of metastatic activity (Bradley et al., 1986) does not affect TEER. Their conditioned media (3T3-CM) did not modify the TEER of bovine umbilical vein endothelial cell (BUVEC-E6E7) monolayers throughout a 6-hour incubation period (new Figure 4e). We have also addressed this issue by showing that conditioned media (CM) from the highly metastatic human breast cancer cell line (MDA-MB-231) (Phannasil et al., 2023) decreased the TEER of primary human umbilical vein endothelial cell (HUVEC) monolayers to a similar level than the CM from B16F10 melanoma cells, whereas the CM from low metastatic human breast cancer cells (MCF7) (Comşa et al., 2015; Phannasil et al., 2023) had no effect (new Figure 4f). Notably, VIAN-c4551 prevented the reduction of the TEER induced by metastatic melanoma and breast cancer cells in HUVEC. These findings support the functional link between increased vasopermeability (TEER values) and metastatic activity, validate the experimental use of BUVEC-E6E7, and extend the potential anti-metastatic influence of VIAN-c4551 to other cancer cell types. The new figures and information have been included in the revised manuscript (new Figures 4e and 4f; lines 83-85; 102, 104-109, 306-320; 332-338; 340; 439-444).

5) In the discussion, in Line 336, the authors stated that VIAN-c4551 inhibits the increase in vascular permeability induced by VEGF by blocking production of NO by eNOS. Is it shown in ref. 19? Please rephrase if it is not a published result.

Reply: The information has been rephrased to clearly indicate corresponding published data. The corrected paragraph (Lines 406-413) is as follows: “Vasoinhibin inhibits the increase in vascular permeability induced by VEGF by promoting the dephosphorylation/inactivation of endothelial nitric oxide synthase (eNOS) leading to the inhibition of the production of nitric oxide (García et al., 2008), an important signal that destabilizes the endothelial cell barrier through posttranslational modifications of adherent junction proteins (Thibeault et al., 2010). VIAN-c4551 also inhibits the VEGF-induced permeability of endothelial cell monolayers (Adán-Castro et al., 2024) and both VIAN-c4551 (Adán-Castro et al., 2024) and vasoinhibin (Arredondo Zamarripa et al., 2014; García et al., 2008) protect against disruption of the blood retinal barrier in experimental models of diabetic retinopathy. However, beneficial effects against the extravasation of premetastatic tumors remain to be determined.”

6) The results show that VIAN-c4551 blocks metastatic cell-induced increase in vascular permeability, presumably through suppressing VEGF action without affecting VEGF expression. So how does this work? As this manuscript does not have data for mechanistic investigation, the authors should at least discuss potential molecular mechanisms such as putative receptors, signaling pathways, distinct features from vasoinhibin etc.

Reply: In attention to the reviewer’s request, we have described potential molecular mechanisms mediating the actions of VIAN-c4551 vs. those of vasoinhibin (Lines 453-461). Description is as follows: “Vasoinhibin binds to a multi-protein complex that includes plasminogen activator inhibitor 1 (PAI-1), urokinase, and the urokinase receptor (Bajou et al., 2014) in endothelial cell membranes which can contribute to the inhibition of multiple signaling pathways (Ras-Raf-MAPK, Ras-Tiam1-Rac1-Pak1, PI3K-Akt-eNOS, and PLCγ-eNOS) activated by several proangiogenic and vasopermeability factors (VEGF, basic fibroblast growth factor, bradykinin, and interleukin-1β) (Clapp et al., 2015). It is unclear how binding to the multi-protein complex inhibits endothelial cells angiogenic processes (Bajou et al., 2014) and the contribution of other vasoinhibin-binding proteins (Clapp & Weiner, 1992) and/or interacting molecules (Morohoshi et al., 2018) likely participate. Along this line, VIAN-c4551 does not bind to PAI-1 (Robles et al., 2023), yet it inhibits VEGF-induced vasopermeability (Adán-Castro et al., 2024; present results) and blood vessel growth (Robles et al., 2022).” The binding molecule/receptor transducing VIAN-c4551’s anti-vasopermeability and antiangiogenic properties are matter of ongoing investigation.

Minor: a number of typos: line 90 - SFB; line 158 - lung paraffin sections from lungs.

Reply: Corrected, thank you (lines 99 and 176-177).

Reviewer 2:

The study titled “The antiangiogenic peptide VIAN-c4551 inhibits lung melanoma metastasis in mice by reducing pulmonary vascular permeability” demonstrates that VIAN-c4551 effectively inhibits lung melanoma metastasis in mice by reducing pulmonary vascular permeability and preventing melanoma cell extravasation. The findings are well-supported by both in vivo and in vitro experiments, and the study opens new avenues for the development of anti-metastatic therapies. However, further research is needed to validate these findings in human clinical trials and to explore the full range of mechanisms by which VIAN-c4551 exerts its anti-metastatic effects.

To further evaluate the study’s findings and address potential limitations, the following questions could be posed to the authors:

Reply: We thank the reviewer for the kind comments and careful evaluation of our work.

Dose Selection and Optimization:

- Why was a single dose of VIAN-c4551 (1 mg/kg) chosen for the experiments? Have you conducted dose-response studies to determine the optimal therapeutic dose?

Reply: A previous study (Robles et al., 2022) showed that VIAN-c4551 (initially named CRIVi45-51) reduced the volume of subcutaneous melanoma tumors in a dose-related manner with 1 mg/kg being the more effective dose. This information has been added to the revised manuscript (Line 142-143).

- Have you explored the pharmacokinetics of VIAN-c4551, such as its half-life, bioavailability, or potential accumulation in tissues, which could influence its efficacy and safety?

- Did you evaluate the potential toxicity of VIAN-c4551 in mice, particularly with long-term administration? Were there any signs of systemic toxicity, such as organ damage or immune suppression?

- Why did you not assess other serum markers (e.g., liver enzymes, kidney function) to evaluate the systemic impact of VIAN-c4551 treatment

Reply: The purpose of our study was to investigate whether, by virtue of its anti-vasopermeability properties, VIAN-c4551 interfered with the extravasation of melanoma cells thereby blocking metastasis generation in mice. Exploring the pharmacokinetics and safety of VIAN-c4551 was beyond the scope of our work. However, in attention to the reviewer’s concern and considering that the findings of our work support the therapeutic potential of VIAN-c4551, we have performed an initial toxicity analysis in six female mice daily injected (i.v.) with vehicle (saline) or with a single dose of VIAN-c4551 (1 or 10 mg/kg) for 14 days. Changes in body weight, blood hematological parameters, serum biochemical parameters, and histopathology were analyzed at the end of the experiment. The two doses of VIAN-c4551 appeared safe as no detrimental effects were observed. These findings are included in the revised manuscript (Tables 1 and 2, Lines 37-39; 42; 199-208; 367-373; 482-484).

In Vitro Model Selection:

- Why did you choose bovine umbilical vein endothelial cells (BUVEC-E6E7) for the in vitro experiments? Did you consider using human endothelial cell lines to better reflect human physiology?

Reply: BUVEC-E6E7 is a well-characterized cell line that maintains the endothelial cell phenotype, including the response to specific regulators (such as VEGF), throughout extended replication times without signs of senescence and genetic instability (Cajero-Juárez et al., 2002). These advantages make BUVEC-E6E7 a good model with repeatable and comparable results. However, in attention to the reviewer’s consideration, we now provide data showing similar findings in primary cultures of human umbilical vein endothelial cells (HUVEC). Like in BUVEC-E6E7, the conditioned media from B16-F10 melanoma cells (B16-F10-CM) decreased the TEER of HUVEC monolayers, VIAN-c4551 prevented such reduction, and VIAN-c4551 alone had no effect (new Figure 4f). Furthermore, VIAN-c4551 prevented the decrease of TEER in HUVEC induced by the CM from a highly metastatic human breast cancer cell line (MDA-MB-231) (Phannasil et al., 2023). These results extend the findings in BUVEC-E6E7 to primary cultures of human endothelial cells (HUVEC) and the potential anti-metastatic influence of VIAN-c4551 to other cancer cell types. (new Figure 4f; lines 89-92; 104-109; 312-320; 334-338).

- Why did you focus on venous endothelial cells rather than arterial endothelial cells, which might be more relevant to the pulmonary vasculature and metastatic seeding?

Reply: Umbilical vein endothelial cells of human origin have been the most frequently used in vitro model in oncology or cardiovascular endothelial cell research (Majewska et al., 2021). Because donor heterogeneity and short-term cultures limit their use, we turned to the immortalized endothelial cell line BUVEC-E6E7. However, we agree with the reviewer that the heterogeneity of endothelial cells in different tissues plays a role in pathophysiology, although the preservation of such differences is challenged in vitro (Majewska et al., 2021). The current study attempted to circumvent in vitro limitations by the in vivo analysis. Nonetheless, the use of lung capillary endothelial cells warrants further research. Part of this information has been added to the revised manuscript (lines 310-315)

References

Adán-Castro, E., Zamora, M., Granados-Carrasco, D., Siqueiros-Márquez, L., García-Rodrigo, J. F., Bertsch, T., Triebel, J., Escalera, G. M. de la, Robles, J. P., & Clapp, C. (2024). Topical Ophthalmic Administration of VIAN-c4551 Antiangiogenic Peptide for Diabetic Macular Edema: Preclinical Efficacy and Ocular Pharmacokinetics (p. 2024.09.11.612517). bioRxiv. https://doi.org/10.1101/2024.09.11.612517

Arredondo Zamarripa, D., Díaz-Lezama, N., Meléndez García, R., Chávez Balderas, J., Adán, N., Ledesma-Colunga, M. G., Arnold, E., Clapp, C., & Thebault, S. (2014). Vasoinhibins regulate the inner and outer blood-retinal barrier and limit retinal oxidative stress. Frontiers in Cellular Neuroscience, 8, 333. https://doi.org/10.3389/fncel.2014.00333

Bajou, K., Herkenne, S., Thijssen, V. L., D’Amico, S., Nguyen, N.-Q.-N., Bouché, A., Tabruyn, S., Srahna, M., Carabin, J.-Y., Nivelles, O., Paques, C., Cornelissen, I., Lion, M., Noel, A., Gils, A., Vinckier, S., Declerck, P. J., Griffioen, A. W., Dewerchin, M., … Struman, I. (2014). PAI-1 mediates the antiangiogenic and profibrinolytic effects of 16K prolactin. Nature Medicine, 20(7), 741–747. https://doi.org/10.1038/nm.3552

Bradley, M. O., Kraynak, A. R., Storer, R. D., & Gibbs, J. B. (1986). Experimental metastasis in nude mice of NIH 3T3 cells containing various ras genes. Proceedings of the National Academy of Sciences of the United States of America, 83(14), 5277–5281. https://doi.org/10.1073/pnas.83.14.5277

Cajero-Juárez, M., Avila, B., Ochoa, A., Garrido-Guerrero, E., Varela-Echavarría, A., Martínez de la Escalera, G., & Clapp, C. (2002). Immortalization of bovine umbilical vein endothelial cells: A model for the study of vascular endothelium. European Journal of Cell Biology, 81(1), 1–8. https://doi.org/10.1078/0171-9335-00213

Clapp, C., Thebault, S., Macotela, Y., Moreno-Carranza, B., Triebel, J., & Martínez de la Escalera, G. (2015). Regulation of blood vessels by prolactin and vasoinhibins. Advances in Experimental Medicine and Biology, 846, 83–95. https://doi.org/10.1007/978-3-319-12114-7_4

Clapp, C., & Weiner, R. I. (1992). A specific, high affinity, saturable binding site for the 16-kilodalton fragment of prolactin on capillary endothelial cells. Endocrinology, 130(3), 1380–1386. https://doi.org/10.1210/endo.130.3.1311239

Comşa, Ş., Cîmpean, A. M., & Raica, M. (2015). The Story of MCF-7 Breast Cancer Cell Line: 40 years of Experience in Research. Anticancer Research, 35(6), 3147–3154.

Dakup, P. P., Porter, K. I., Little, A. A., Zhang, H., & Gaddameedhi, S. (2020). Sex differences in the association between tumor growth and T cell response in a melanoma mouse model. Cancer Immunology, Immunotherapy : CII, 69(10), 2157–2162. https://doi.org/10.1007/s00262-020-02643-3

García, C., Aranda, J., Arnold, E., Thébault, S., Macotela, Y., López-Casillas, F., Mendoza, V., Quiroz-Mercado, H., Hernández-Montiel, H. L., Lin, S.-H., Escalera, G. M. de la, & Clapp, C. (2008). Vasoinhibins prevent retinal vasopermeability associated with diabetic retinopathy in rats via protein phosphatase 2A–dependent eNOS inactivation. The Journal of Clinical Investigation, 118(6), 2291–2300. https://doi.org/10.1172/JCI34508

Majewska, A., Wilkus, K., Brodaczewska, K., & Kieda, C. (2021). Endothelial Cells as Tools to Model Tissue Microenvironment in Hypoxia-Dependent Pathologies. International Journal of Molecular Scienc

---

## [Editor Report · Decision Letter 1]

8 Apr 2025

The antiangiogenic peptide VIAN-c4551 inhibits lung melanoma metastasis in mice by reducing pulmonary vascular permeability

PONE-D-24-58885R1

Dear Dr. Clapp,

We’re pleased to inform you that your manuscript has been judged scientifically suitable for publication and will be formally accepted for publication once it meets all outstanding technical requirements.

Kind regards,

Mohamed Abdelkarim

Academic Editor

PLOS ONE
---

## [Editor Report · Acceptance letter]

PONE-D-24-58885R1

PLOS ONE

Dear Dr. Clapp,

I'm pleased to inform you that your manuscript has been deemed suitable for publication in PLOS ONE. Congratulations! Your manuscript is now being handed over to our production team.

Kind regards,

on behalf of

Dr. Mohamed Abdelkarim

Academic Editor

PLOS ONE